# Adherence to Pulmonary Tuberculosis Medication and Associated Factors Among Adults: A Cross-Sectional Study in the Metinaro and Becora Sub-Districts, Dili, Timor-Leste

**DOI:** 10.3390/ijerph21121662

**Published:** 2024-12-13

**Authors:** Amentinho Fernandes, Sawanya Laohaprapanon, Truong Thanh Nam, Ercia Maria Da Conceicao Sequeira, Cua Ngoc Le

**Affiliations:** 1Environmental Safety Technology and Health Program, School of Public Health, Walailak University, Nakhon Si Thammarat 10860, Thailand; amentinho.fe@mail.wu.ac.th (A.F.); sawanya.la@mail.wu.ac.th (S.L.); 2National Hospital Guido Valadares, Dili 670001, Timor-Leste; daconceicaosequeirae@gmail.com; 3Center of Excellence in Data Science for Health Study, Walailak University, Thasala District, Nakhorn Si Thammarat 10860, Thailand; 4Public Health Research Program, School of Public Health, Walailak University, Nakhon Si Thammarat 10860, Thailand; ttnam@ctump.edu.vn; 5Faculty of Public Health, Can Tho University of Medicine and Pharmacy, Can Tho City 94000, Vietnam; 6Department of Infectious Disease, National Hospital Guido Valadares, Dili 670001, Timor-Leste; 7Excellent Center for Public Health Research (ECPHR), Walailak University, Nakhon Si Thammarat 80160, Thailand

**Keywords:** pulmonary tuberculosis, medication adherence, influence factors, Timor-Leste

## Abstract

Timor Leste is one of the top countries in Asia with a high incidence rate of pulmonary tuberculosis (TB). The success of TB treatment necessitated a more profound comprehension of adherence as a multifaceted behavioral issue, along with identifying the barriers that hinder and the factors that promote patient adherence. This study aimed to assess the rate of pulmonary TB medication adherence and identify its predictors among adults in Metinaro and Becora, Dili, Timor-Leste. A descriptive analytical cross-sectional study was conducted, and new patients with pulmonary TB aged 18 years and above were selected using a proportional sampling method. Quantitative data were collected from 398 patients with pulmonary tuberculosis. The medication adherence results were as follows: 73.6% low adherence, 18.3% moderate adherence, and only 8.1% high adherence. The study identified significant predictors of medication adherence, such as health service factors (OR = 14.024, 95% CI: 5.42–35.54, *p* = 0.001). Patients who perceived a high quality in the health service were 14 times more likely to exhibit higher medication adherence. Regarding individual behaviors, patients who consumed alcohol or occasionally engaged in physical exercise were significantly less likely to exhibit higher medication adherence (OR = 0.17, 95% CI: 0.091–0.312, *p* = 0.001). Similarly, patients experiencing high levels of stigma were less likely to achieve strong adherence (OR = 0.146, 95% CI: 0.058–0.326, *p* = 0.001).Both health service quality and individual factors, such as lifestyle behaviors and social stigma, were statistically significant predictors ofTB medication adherence. Enhancing the healthcare infrastructure, implementing multisectoral strategies for behavior change, and reducing stigma are crucial. Additionally, mobile health technologies, like SMS reminders and telehealth, might support real-time adherence improvements.

## 1. Background

Tuberculosis (TB) is the second leading infectious killer after COVID-19. In 2022, an estimated 10.6 million incident TB cases and 1.4 million TB-related deaths were reported [1]. According to the World Health Organization (WHO) Global TB Report 2022, Timor-Leste has one of the highest TB burdens in Southeast Asia, with an incidence rate of 498 per 100,000, making it the seventh highest in the world, and a mortality rate of 106 per 100,000, the highest globally [1,2]. TB is also the eighth leading cause of death in Timor-Leste [2]. The disease predominantly affects individuals aged 15–34 years, with a concerning prevalence of multidrug-resistant tuberculosis (MDR-TB)/rifampicin-resistant tuberculosis (R-R TB) of 2.5% among new cases and 14% among retreatment cases [3].

The standard TB treatment regimen recommended by WHO requires at least six months, comprising an intensive phase of two months with a four-drug regimen (isoniazid, rifampicin, pyrazinamide, and ethambutol), followed by a continuation phase lasting more than four months [1,4] Adherence to this regimen is challenging, particularly during the continuation phase when symptoms were improved, leading to a dropout rate of 40% in medication adherence [5]. Studies have shown that patients often become careless with their medication once they start feeling better, mistakenly believing they are cured [6,7]. Severe consequences of poor medication adherence include drug resistance, recurrence, and disease deterioration. It also makes Mycobacterium tuberculosis (MTB) more contagious, which adds to the difficulty of TB control [8].

Several interrelated factors affected TB medication adherence, including a lack of knowledge about TB and its treatment regimen, healthcare accessibility issues, and social stigma [7]. Additional challenges included transportation barriers, loss of employment, adverse medication effects, long treatment periods, and poor communication with healthcare providers) [9]. Direct observation of therapy (DOT) remains a significant challenge for TB control programs in many developing countries, including Timor-Leste, despite the near-universal implementation of a Direct Observation Treatment Short (DOTS) Course in most healthcare facilities [10]. Local strategies and the decentralization of TB services have been proven to improve treatment success rates because treatment completion remains the biggest challenge in global TB control (Martins) [11].

Despite these challenges, addressing the determinants of TB medication adherence is crucial for improving treatment outcomes. Implementing effective local strategies and decentralizing TB treatment services have improved treatment success rates. However, the health authorities in Timor-Leste should make sure that directly observed therapy (DOT) is applied consistently. Therefore, we conducted the study in the high-incidence sub-districts of Metinaro and Becora to evaluate pulmonary tuberculosis medication adherence and find the associated factors. The study results might improve TB medication adherence by tackling these factors.

## 2. Methodology

### 2.1. Study Design and Setting

This cross-sectional survey was conducted between August and December 2023 in the community health centers of the Becora and Metinaro sub-districts in the city of Dili. These sub-districts, located on the northern coast of Timor, are key areas for studying TB due to their high incidence rates. The study targeted adult patients in the continuation phase of TB treatment.

### 2.2. Participant Selection and Sampling

Eligible participants for this study were new patients with pulmonary tuberculosis (PTB) receiving treatment at two community health centers. They were required to be aged 18 years or older and have completed at least 3 months of treatment but were yet to reach the complete 6-month treatment regimen. We excluded those who had been undergoing treatment for less than 3 months, were being treated for multidrug-resistant TB (MDR-TB) or extensively drug-resistant TB (XDR-TB), had extrapulmonary TB, were unable to complete the survey, or were undergoing psychotherapy.

The estimated sample size was calculated using a single population proportion formula with an assumed adherence rate of 62% [12], a Z score of 1.96 at a 95% confidence level, and a margin of error of 5%, resulting in a sample size of 362 patients with TB. To account for a 10% non-response rate, ensuring a response rate of 90%, the final sample size was adjusted to 398 participants. Finally, we received feedback from all 398 participants, achieving a 100% response. A stratified random sampling or proportional sampling was employed to select the participants. Both the Metinaro and Becora sub-districts in the Dili district had a total of 613 patients with pulmonary tuberculosis (PTB). The sample size of 398 participants represented 64% of the total PTB population. Todetermine the sample size for each sub-district, the number of patients with PTB in Metinaro (392) and Becora (221) was multiplied by 64%, resulting in proportional sample sizes for each sub-district. Participants were stratified based on medical registration, assigned numerical values, and randomly selected using Microsoft Excel to ensure a representative and unbiased sample.

### 2.3. Research Instrument

The data for this research were collected using a structured questionnaire adapted from a previous study [7]. The questionnaire consisted of eight sections, designed to gather comprehensive information on the various factors influencing TB medication adherence (Appendix A):

Section 1: Sociodemographic characteristics: This section included eight questions to capture data on gender, age, marital status, educational level, occupation, monthly income, and living area. These factors provided a baseline understanding of the participants’ backgrounds.

Section 2: Knowledge of patients on tuberculosis, transmission, and treatment control: This part included six multiple-choice questions to assess knowledge about causing agents, TB symptoms, TB transmission, and TB treatments. Each correct answer scored 1, and each incorrect response scored 0, resulting in a total score out of 6 points. A cut-off point of 50% was used to classify the respondents’ knowledge as poor (<50%) or good (≥50%) [6].

Sections 3, 6, and 7: “Health service qualities” and “Stigma and family support” included five items and four items, respectively. Patients rated these items based on a 5-point Likert scale varying in a range: strongly disagree (1 point), disagree (2 points), neutral (3 points), agree (4 points), and strongly agree (5 points). Health service-related factors provided insight into their accessibility and affordability of TB services, satisfaction, confidence, and staff support. Stigma and family support measured feelings of shame and stigma, and support from their family and community. The Stigma and family support analysis used predefined cut-off points [13,14].

Section 4: Health condition-related factors included four questions to understand patient’s different health conditions based on multiple-choice questions.

Section 5: Behavioral factors evaluated physical exercise, alcohol consumption, and tobacco use. Respondents were asked about their frequency of engaging in these activities, with the options categorized as daily, occasional, or never. Responses were coded as 1 for daily/occasional and 0 for never [15].

Section 8: TB medication adherence: Medication adherence was measured using the 8-item Morisky Medication Adherence Scale (MMAS-8). Patients were classified into three groups based on their scores: high adherence (score of 8), medium adherence (score of 6–7), and low adherence (score < 6) [16]. The MMAS-8 has demonstrated acceptable reliability and validity, with a Cronbach’s α of 0.74 in this study.

The quality of the measurement was ensured by adopting well-validated contents and the reliability of the scales. Three experts assessed the content validity of the items from Sections 2 to 7 in the questionnaire based on the degree of relevance of each item to the measured domains according to the 4-pointLikert scales, as follows: (1) Not relevant, (2) Somewhat relevant, (3) Quite relevant, and (4) Very relevant. The calculation of the item-level of CVI (I-CVI) included the number of experts giving 3 or 4 points for an item divided by the total number of experts [17]. The average of the scale level of the CVI (S-CVI/Ave) was calculated by the average of the I-CVI of the items. The universal agreement of the scale level of the CVI (S-CVI/UA)is defined as the percentage of items on the scale that are given a relevant scale of 3 or 4 by all the experts. The acceptable S-CVI/Ave and S-CVI/UA of the research questionnaire should be at least 0.80 [18].

For the reliability test of the questionnaire, three sections applying Likert scales such as “Health service qualities”, “Stigma and family support”, and “Community support” were assessed among 30 patients with TB in the pilot study. Those patients with TB were not chosen in the sample size of the research. Cronbach’s alpha was calculated to determine the internal consistency of each section. A Cronbach’s alpha of 0.7 or higher was considered acceptable, indicating good reliability [19].

### 2.4. Data Collection

After obtaining ethical approval, the researcher recruited two senior nurses and two public health specialists in tropical disease control working in community health centers to assist with the data collection. The researcher organized two online meetings to explain the study’s objectives, ethical guidelines, tools, survey procedures, and participant selection.

The data collectors approached potential participants, explained the study’s purpose, obtained informed consent, and clarified questionnaire items if they were difficult to comprehend. Participants were contacted by phone or in person, and research assistants selected replacements if they declined. We maintained confidentiality by encoding personal identifiers and storing the completed survey data securely.

### 2.5. Data Analysis

Firstly, a descriptive study assessed the frequency distributions, means, medians, standard deviations, and interquartile ranges, providing a baseline understanding of the data. Secondly, a two-stage statistical approach(chi-square tests followed by ordinal logistic regression) was used: Stage 1: Chi-square tests screened for variables that have a statistically significant relationship with medication adherence; Stage 2: Ordinal logistic regression incorporated significant variables (*p* < 0.25) identified from the chi-square tests, to determine and quantify the predictors of medication adherence [20]. All the statistical analyses were performed using the SPSS software, version 22 (IBM Corp., Armonk, NY, USA). A *p*-value of less than 0.05 was considered statistically significant.

### 2.6. Ethics Approval and Informed Consent

The study protocol was reviewed and approved by the Human Research Ethics Committee at Walailak University, in compliance with the ethical principles for medical research involving human subjects as outlined in the Helsinki Declaration. This approval is documented under the Ethics Committee reference number WUEC-23-247-01.

Prior to participation, all the subjects received verbal and written information about the purpose and structure of the study. Participants provided written informed consent, ensuring that they understood their involvement and rights. The study emphasized informed consent, participant welfare, and adherence to ethical guidelines throughout the research process.

## 3. Results

### 3.1. Content Validity and Reliability of the Questionnaire

Three experts gave each item a score between three and four points, with 19 out of 22 items in six different questionnaire parts receiving a score of 1. According to Yusoff (2019), the S-CVI/Ave and S-CVI/UA content validity indices were 0.95 and 0.86, respectively, indicating satisfactory content validity levels. The questionnaire also showed acceptable internal consistency with Cronbach’s alpha values of 0.78 for the Health service-related factors and Community support subscales and 0.7 for the Stigma and family support subscale, as presented in Table 1.These findings suggested that the questionnaire was appropriate for the primary study and had sufficient reliability (see Table 1).

### 3.2. Socio-Demographic Characteristics of Research Participants

The results of the medication adherence assessment among the 398 patients with pulmonary tuberculosis showed that 293 (73.6%), 73 (18.3%), and 32 (8.1%) patients achieved low, medium, and high adherence, respectively. The screening stage by the Chi-squared test resulted in variables such as gender, age, marital status, and occupation, with *p*-values below 0.25 that were included in the ordinal logistic regression analysis for predicting the TB medication adherence in the second stage, as detailed in Table 2.

### 3.3. Association Between TB Medication Adherence and Knowledge, Health Service Factors, Health Condition Factors, Patient Behavior Factors, Stigma, and Family Support of Participants

The results in Table 3 indicated that health service quality was consistently assessed as ranging from moderate to high, with no participants reporting low-quality services. None of the participants had a concurrent HIV infection. Participants predominantly reported low levels of family and community support from their perspectives.

After screening for variables by Chi-square test, variables such as TB knowledge, health service quality, medication side effects, smoking habit, alcohol consumption, physical exercises, and perceived stigma were included in the ordinal logistic regression model to determine the predictors of TB medicine adherence.

### 3.4. Predictors of Patients’ Medication Adherence Based on Ordinal Logistic Regression

In Table 4, the ordinal logistic regression analysis identified health service quality, alcohol consumption, physical exercise, and perceived stigma to be significant predictors of medication adherence among patients with TB in the Metinaro and Becora sub-districts of Dili, Timor-Leste. The odds ratio (OR) for “Health service quality” was reported as 14.024. This indicated that patients who perceived the health service as high quality were 14 times more likely to exhibit higher medication adherence compared with those who perceived it as moderate quality (OR = 14.024, 95% CI: 5.42–35.54, *p* = 0.0001). For patients’ habits such as alcohol consumption and physical exercise, patients who consumed alcohol or occasionally performed physical exercise were significantly less likely to demonstrate higher medication adherence than non-drinkers or daily exercise performers (OR = 0.17, 95% CI: 0.091–0.312, *p* = 0.0001). In this context, an OR of0.17 implied that alcohol consumption and occasional physical exercise were significantly associated with lower medication adherence among patients. Finally, patients experiencing high levels of stigma were less likely to achieve higher medication adherence compared with those perceiving moderate stigma (OR = 0.146, 95% CI: 0.058–0.326, *p* = 0.0001).

Conversely, certain variables that did not show statistically significant predictors of medication adherence included gender, age, marital status, occupation, knowledge, experience of side effects, and smoking habits (Table 4).

## 4. Discussion

In the current study, we used the MMAS-8 to investigate TB medical adherence among 398 patients with pulmonary TB in Dili, Timor-Leste and found that 73.6%, 18.3%, and 8.1% of patients demonstrated low, moderate, and high adherence to TB medication, respectively. To ensure a consistent approach to assessing adherence to TB medication, we compared the findings to the Chinese studies using the Morisky Medication Adherence Scale (MMAS-8). In this study, 73.6% of patients with TB demonstrated low adherence, which is considerably higher than the 34.6% of low adherence reported among rural patients with TB in China [16] and 25.71% of low adherence among patients with TB in Dalian, Northeast China [7]. This discrepancy could be attributed to differences in healthcare infrastructures, social support, stigma, and other socio-cultural factors between Dili, Timor-Leste and the regions in China. These significant differences highlight the need for tailored interventions to increase TB medicine adherence in Dili, Timor-Leste based on the local context.

Regarding significant predictors of TB medicine adherence, health service quality, alcohol consumption, physical exercise, and perceived stigma were identified as significant predictors of medication adherence. Patients perceiving high-quality health services were 14 times more likely to adhere to TB medication. Previous studies showed that systemic issues like prescription stockouts and limitations with drug transportation to a transient community have resulted in the limited availability of anti-tuberculosis medications [10,21]. The Chinese study identified treatment-related factors such as following doctors’ advice and using adjuvant drugs as essential determinants of adherence [7]. These concepts are similar to the satisfaction of doctor support, quality of care, support service, and the availability and easy access to medicine included in health service quality. These findings emphasize the role of healthcare quality in adherence, consistent with the literature that associates better healthcare delivery with improved patient compliance. Additionally, our study found that alcohol consumption was a significant barrier to TB medicine adherence, which aligns with the existing literature across various chronic diseases. Substance use, particularly alcohol consumption, has been consistently associated with lower medication adherence. Du et al. also revealed that patients with TB in Dalian, China who did not consume alcohol were more likely to have higher medicine adherence, with an OR of 1.84 (*p* = 0.032) [7]. Besides TB, alcohol consumption caused the negative impact on medication adherence across various diseases. In chronic diseases like hypertension and diabetes, alcohol consumption has been negatively associated with medication adherence [22]. These findings suggest that substance use consistently acts as a barrier across different cultural settings and underscores the importance of addressing substance use behaviors to enhance medication adherence across diverse patient populations. Regarding the second lifestyle behavior, the current study released that patients who engaged in occasional physical exercise were significantly less likely to demonstrate higher medication adherence than daily exercise performers. This finding might reflect a general lack of routine, discipline, or commitment, which could extend to low medication adherence. Our finding aligns with the result of the study in China, indicating that a lack of regular physical activity is linked to lower proactive health behaviors, such as seeking medical help or adhering to prescribed medication [15]. Individuals who are less active may generally exhibit lower engagement in self-care and preventive behaviors. The association between physical activity and health behaviors suggests that routine exercise promotion could have broader benefits, improving not only physical fitness but also medication adherence and healthcare utilization.

Finally, our study revealed that a high perceived stigma was significantly associated with low medicine adherence. Other studies also concluded that perceived stigma seems to be related to lower medication adherence and an increased risk of tuberculosis [7,16,23]. The findings in these studies conclude stigma as a significant barrier to medication adherence. This underscores the importance of interventions focused on reducing stigma to improve medication adherence rates.

In this study, several factors commonly believed to influence TB medication adherence, such as gender, age, and knowledge about TB, were not found to be significantly associated with medication adherence levels among the current study population.

A prospective observational study conducted in India showed that the incidence of TB was higher in males [23]. The Chinese study found that increasing age was significantly associated with better medication adherence, with an odds ratio (OR) of 1.02 per year (*p* = 0.013). While statistically significant, this OR suggests only a marginal effect size, indicating that age alone may not strongly impact adherence. The association may instead reflect underlying factors such as greater health awareness, life experience, or stronger beliefs in following medical advice [7]. In addition, previous studies have emphasized that patient knowledge is a crucial factor in determining adherence to TB treatment [9,24,25]. Others revealed that the presence of comorbidities has been noted to contribute to non-adherence to TB medication [26,27].

The above result suggests that while these variables often play a crucial role in shaping adherence behaviors in other contexts, they may not exert the same influence in the specific setting of Dili, Timor-Leste. Understanding why these factors did not reach statistical significance requires a closer examination of local socio-cultural dynamics, healthcare delivery, and patient perceptions that might overshadow the expected impact of these variables. For instance, knowledge about TB was not significantly associated with medication adherence in our study, which contrasts with the findings from another research. From the perspective of healthcare delivery, this might explain that health education regarding TB prevention and treatment in Timor-Leste remains inadequate [11].

Additionally, the changes in the odds ratios (ORs) from univariate to multivariate analyses, as observed for knowledge of TB, highlight the role of confounding. In the univariate analysis, knowledge of TB showed a significant association with adherence; however, in the multivariate model, the OR became 1.001 (~1) and was no longer significant. This shift suggests that the initial association observed in the univariate analysis was confounded by other factors such as health service quality or perceived stigma. These variables may have had independent effects on adherence and, when taken into account in the multivariate model, decreased the apparent impact of knowledge on adherence. This result emphasizes how TB drug adherence is multifaceted and how crucial it is to control for confounders in order to identify the distinct contributions of each variable. Understanding these dynamics emphasizes the need for tailored interventions addressing both individual knowledge and broader systemic and social barriers to adherence.

Despite the contributions to understanding the issue, we acknowledge the following limitations that might affect the interpretation and generalizability of the findings and provide directions for future research. First, the study utilized self-reported data through the Morisky Medication Adherence Scale, which might be influenced by social desirability and recall biases. Second, the study’s findings, based on data from the Metinaro and Becora sub-districts of Dili, Timor-Leste, might have limited generalizability to other regions with different healthcare systems, socio-cultural contexts, or patient demographics. Next, the cross-sectional design of the study might limit the ability to determine causal relationships between predictors and medication adherence. Finally, despite using a stratified random sampling approach, potential selection bias might have occurred if patients who were more accessible or willing to participate had different adherence behaviors compared with those not included in the study.

## 5. Conclusions

In this study, the rate of TB medication adherence in the Metinaro and Becora sub-districts of Dili, Timor-Leste, revealed that 73.6% of patients demonstrated low adherence. Such a high prevalence of low adherence emphasizes an urgent need for targeted interventions to address the barriers contributing to poor adherence. Among the significant predictors of TB medicine adherence, health service quality emerged as the most significant predictor of medication adherence. Meanwhile, alcohol consumption and occasional physical exercise were considered as barriers to consistent medication. Perceived stigma further contributed to lower adherence rates, underscoring the impact of social perceptions on medicine adherence. Overall, both structural factors like healthcare quality and individual-level factors like behaviors and social stigma play critical roles in shaping medication adherence among patients with TB in Timor-Leste.

The following actions and policies might address these above factors. Firstly, improving health service quality might address both systemic and patient-centered barriers through better infrastructure, patient-centered communication, and consistent service delivery. A multifaceted approach involving healthcare, health counseling, and community engagement addresses the determinants of adherence, such as the reduction of alcohol consumption, promotion of regular physical activity, and mitigation of stigma. Incorporating mobile health technologies, such as SMS reminders and telehealth, can enhance effectiveness by enabling healthcare providers to monitor adherence remotely and provide timely interventions for patients at risk of non-adherence. These combined strategies can create a supportive healthcare environment that addresses psychological and informational barriers to enhance TB medication adherence.

## Figures and Tables

**Table 1 ijerph-21-01662-t001:** Content validity of items in the questionnaire from part 2 to part 7.

Items/Questionnaire	Expert 1	Expert 2	Expert 3	I-CVI
Part 2. Knowledge of patients on tuberculosis, transmission, and treatment control				
K1	3	4	4	1
K2	4	4	3	1
K3	4	4	4	1
K4	4	4	4	1
K5	4	4	4	1
K6	4	3	2	0.66
Part 3. Health service-related factors				
HSF1	4	4	3	1
HSF2	4	2	4	0.66
HSF3	4	3	2	0.66
HSF4	4	4	3	1
HSF5	4	4	3	1
Part 4. Health condition-related factors				
HCF1	3	4	4	1
HCF2	4	4	3	1
HCF3	4	4	4	1
HCF4	4	4	4	1
Part 5. Patients’ behavioral factors				
PBF1	3	4	4	1
PBF2	4	4	3	1
PBF3	4	4	4	1
Part 6. Stigma and family support				
SF1	4	4	3	1
SF2	4	4	3	1
Part 7. Community support				
FC1	4	4	3	1
FC2	4	4	4	1

Note: S-CVI/Ave = 0.95; S-CVI/UA = 0.86.

**Table 2 ijerph-21-01662-t002:** Characteristic of study participants related to medication adherence levels.

Variable	N (%)	Medication Adherence Level			Chi-Square	*p*-Value
		Low	Medium	High		
Total	398 (100%)	293(73.6)	73 (18.3)	32 (8.1)		
Gender						
Male	217 (54.5)	166 (76.5)	40 (18.4)	11 (5.1)	5.778	0.056
Female	181 (45.5)	127 (70.2)	33 (18.2)	21 (11.6)		
Age						
18–30	115 (28.9)	73 (63.5)	26 (22.6)	16 (13.9)	12.575	0.050
31–50	112 (28.1)	88 (78.6)	19 (17.0)	5 (4.5)		
51–64	100 (25.1)	74 (74.0)	18 (18.0)	8 (8.0)		
>65	71 (17.8)	58 (81.7)	10 (14.1)	3 (4.2)		
Marital status						
Single	100 (25.1)	65(65)	26 (26)	9 (9)	7.052	0.133
Married	245 (61.6)	188 (76.7)	40 (16.3)	17 (6.9)		
Divorced	53 (13.3)	40 (75.5)	7 (13.2)	6 (11.3)		
Education level						
Non formal	48 (12.1)	36 (75.0)	8 (16.7)	4 (8.3)	4.642	0.590
Primary school	144 (36.2)	110 (76.4)	25 (17.4)	9 (6.2)		
Secondary school	203 (51.0)	146 (71.9)	39 (19.2)	18 (8.9)		
Tertiary school	3 (0.8)	1 (33.3)	1 (33.3)	1 (33.3)		
Occupation						
Unemployed	56 (14.1)	38 (67.9)	14 (25.0)	4 (7.1)	13.743	0.089
Farmer	141 (35.4)	111 (78.7)	23 (16.3)	7 (5.0)		
Officer	1 (0.3)	1 (100.0)	0 (0.0)	0 (0.0)		
Merchant	18 (4.5)	8 (44.4)	7 (38.9)	3 (16.7)		
Laborer	182 (45.7)	135 (74.2)	29 (15.9)	18 (9.9)		
Monthly income						
<150$	387 (97.2)	286 (73.9)	70 (18.1)	31 (8.0)	1.417	0.841
150–500$	10 (2.5)	6 (60.0)	3 (30.0)	1 (10.0)		
250–500$	1 (0.3)	1 (100.0)	0 (0.0)	0 (0.0)		
Living area						
Urban	80 (20.1)	60 (75.0)	12 (15.0)	8 (10.0)	3.865	0.425
Rural	289 (72.6)	208 (72.0)	58 (20.1)	23 (8.0)		
Remote	29 (7.3)	25 (86.2)	3 (10.3)	1 (3.4)		

**Table 3 ijerph-21-01662-t003:** Association between pulmonary TB medication adherence and knowledge, health services factor, health conditions factors, health behavior, stigma, and family support.

Variable	N (%)	Medication Adherence Level			Chi-Square	*p*-Value
		Low	Medium	High		
TB knowledge						
Poor	243 (61.1)	179 (73.7)	36 (14.8)	28 (11.5)	13.643	0.001
Good	155 (38.9)	114 (73.5)	37 (23.9)	4 (2.6)		
Health service quality						
Moderate	33 (8.3)	7 (21.2)	26 (78.8)	0 (0.0)	87.998	0.001
High	365 (91.7)	286 (78.4)	47 (12.9)	32 (8.8)		
Chronic health conditions						
Absence of CHC	358 (89.9)	267 (74.6)	65 (18.2)	26 (7.3)		
DM	6 (1.5)	4 (66.7)	1 (16.7)	1 (16.7)		
HIV/AIDS	-	-	-	-	3.264	0.515
Others (Asthma, COPD)	34 (8.5)	22 (64.7)	7 (20.6)	5 (14.7)		
Period of treatment						
3 months	95 (38.3)	68(71.6)	18 (9.6)	9 (4.8)		
4 months	161 (32.8)	122 (75.8)	31 (19.3)	8 (5.0)		
5 months	75 (15.3)	55 (73.3)	12 (16)	8 (10.7)	3.773	0.707
6 months	67 (13.6)	48 (71.6)	12 (17.9)	7 (10.4)		
TB medication side effects						
Reported	107 (26.9)	78 (72.9)	7 (6.5)	22 (20.6)	45.744	0.001
Nausea	61 (15.3)	47 (77)	13 (21.3)	1 (1.6)		
Headache	31 (7.8)	19 (61.3)	11 (35.5)	1 (3.2)		
Dizziness	79 (19.8)	57 (72.2)	20 (25.3)	2 (2.5)		
Gastrointestinal disturbance	52 (13.1)	40 (76.9)	9 (17.3)	3 (5.8)		
Change in sleep	68 (17.1)	52 (76.5)	13 (19.1)	3 (4.4)		
Physical exercise						
Experienced	368 (92.5)	276 (75)	64 (17.4)	28 (7.6)	4.801	0.091
None	30 (7.5)	17 (56.7)	9 (30)	4 (13.3)		
Alcohol consumption						
Experienced	368 (92.5)	276 (75)	64 (17.4)	28 (7.6)	4.801	0.091
None	30 (7.5)	17 (56.7)	9 (30)	4 (13.3)		
Smoking habit						
Experienced	209 (52.5)	165 (78.9)	38 (18.2)	6 (2.9)	19.8	0.001
None	189 (47.5)	125 (66.1)	35 (18.5)	29 (15.3)		
Stigma						
Moderate	229 (57.5%)	184 (80.3)	18 (7.9)	27 (11.8)	45.055	0.001
High	169 (42.5%)	109 (64.5)	55(32.5)	5 (3.0)		
Family and community support						
Low	398	293 (73.6)	73 (18.3)	32 (8.1)		1

Note: Chronic health condition (CHC); Chronic obstruction pulmonary disease (COPD); Diabetes mellitus (DM).

**Table 4 ijerph-21-01662-t004:** Predictors of medication adherence based on a multiple ordinal logistic regression.

Determinants of Patients with Pulmonary TB		B	SE	Wald	df	OR	Sig.	95% Confidential Interval	
								Lower	Upper
Gender	Male	−0.164	0.255	0.414	1	0.848	0.52	0.51	1.39
	Female	Ref	-	-	0	-	-	-	-
Age	18–30	0.473	0.546	0.751	1	1.605	0.386	0.55	6.77
	31–50	0.331	0.508	0.426	1	1.39	0.514	0.52	3.76
	51–64	0.270	0.534	0.255	1	1.310	0.614	0.46	3.73
	>65	Ref		-	0	-	-	-	-
Marital status	Single	0.126	0.434	0.085	1	0.881	0.771	0.48	2.65
	Married	−0.351	0.394	0.796	1	0.704	0.372	0.32	1.52
	Divorced	Ref	-	-	0	-	-	-	-
Occupation	Farmer	−0.179	0.452	0.156	1	0.836	0.996	0.34	2.65
	Officer	−16.798	0.000	3.242	1	5.10^−8^	0.692	5.10^−8^	5.10^−8^
	Merchant	1.086	0.603	0.003		2.963	0.072	1.101	9.65
	Laborer	−0.022	0.385			0.978	0.954	0.460	2.08
	Unemployed	Ref	-	-	0	-	-	-	-
Knowledge	Poor	0.001	0.306	0.000	1	1.001	0.996	0.551	1.82
	Good	Ref	-	-	0	-	-	-	-
Health service quality	High	2.639	0.480	30.232	1	14.024	0.000	5.42	35.54
	Moderate	Ref	-	-	0	-	-	-	-
Medication side effect	Yes	−0.613	0.391	2.466	1	0.541	0.116	0.68	1.164
	No	Ref	-	-	0		-	-	-
Smoking habit	Yes	−0.843	1.23	0.471	1	0.431	0.492	0.039	4.73
	No	Ref	-	-	0	-	-	-	-
Alcohol consumption	Yes	−1.783	0.313	32.49	1	0.17	0.000	0.091	0.312
	No	Ref	-	-	0	-	-	-	-
Physical exercise	Never	−0.843	1.22	0.471	1	0.43	0.49	−3.25	1.56
	Occasionally	−1.783	0.31	32.49	1	0.17	0.000	0.091	0.312
	Daily	Ref	-	-	0	-	-	-	-
Perceive stigma	High	−1.927	0.412	21.88	1	0.146	0.000	0.058	0.326
	Moderate	Ref	-	-	0	-	-	-	-

Note: Ref: Reference; OR: odds ratio; B: Beta coefficient; SE: standard error.

## Data Availability

The original contributions presented in this study are included in the article/Appendix A. Further inquiries can be directed to the corresponding author.

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
