# Peer review of "Adherence to Pulmonary Tuberculosis Medication and Associated Factors Among Adults: A Cross-Sectional Study in the Metinaro and Becora Sub-Districts, Dili, Timor-Leste"

_ijerph, 2024, doi:10.3390/ijerph21121662_

Round 1
Reviewer 1 Report (Previous Reviewer 1)
Comments and Suggestions for Authors
This is a revision of a paper that was apparently withdrawn after reviewers’ comments, but resubmitted.
The manuscript has improved substantially and I think the authors all together have followed my comments and changed the paper accordingly.
I now find the paper of sufficient quality without major remaining issues that need to be addressed. However, there are still a few issues:
First – I cannot see how 'low, medium and high adherence to treatment' are defined. I may have missed it, but it is highly important that this is defined in the text.
Second – page 14, line 463 – an OR of 1.02 (quotation from a Chinese study), is that per year? To the reader this should be defined, as an OR as low as 1.02, although significant, is so close to 1 that is has no biological importance, unless this is e.g. change in OR per year.
Third – in the discussion of results from the multiple ordinal logistic regression model, the concept of confounding is not mentioned at all. This is what a regression model does – adjust for confounders. Thus, a factor like knowledge of TB that was clearly associated with adherence to treatment in the univariate analyses (Table 3), was not significant in the regression model with an OR of 1.001 (~1). I think it should be discussed that changes in OR of the different factors from the univariate analyses to the multivariate analysis could be ascribed to confounding, and in case in what direction. Following that there is too much emphasis on statistical significance (e.g. the section from line 485 page 15) rather than actual changes in OR and the possibility of confounding.
Author Response
First – I cannot see how 'low, medium and high adherence to treatment' are defined. I may have missed it, but it is highly important that this is defined in the text.
Response: We appreciate the opportunity to clarify this. The classification of "low, medium, and high adherence to treatment" has been defined in the manuscript under Section 8: TB Medication Adherence of the Research Instrument/Method section. Specifically, medication adherence was measured using the eight-item Morisky Medication Adherence Scale (MMAS-8), with the following categorization:
High adherence: Score of 8
Medium adherence: Score of 6–7
Low adherence: Score <6
This classification follows established thresholds used in previous research, such as Xu et al. (2017), and ensures consistency in the interpretation of adherence levels. We trust this explanation addresses your concern and confirms its inclusion in the manuscript.
Second – page 14, line 463 – an OR of 1.02 (quotation from a Chinese study), is that per year? To the reader this should be defined, as an OR as low as 1.02, although significant, is so close to 1 that is has no biological importance, unless this is e.g. change in OR per year.
Response: Thank you for highlighting this point. The odds ratio (OR) of 1.02 reported in the Chinese study refers to the change in medication adherence odds per one-year increase in age. While this OR is statistically significant, we agree that its biological or clinical relevance should be interpreted cautiously due to its proximity to 1. This suggests that age alone may not have a substantial independent impact on adherence behaviors. Instead, it might reflect other underlying factors associated with increasing age, such as greater health awareness or stronger beliefs in following medical advice. We have revised like this "The Chinese study found that increasing age was significantly associated with better medication adherence, with an odds ratio (OR) of 1.02 per year (p = 0.013). While statistically significant, this OR suggests only a marginal effect size, indicating that age alone may not strongly impact adherence. The association may instead reflect underlying factors such as greater health awareness, life experience, or stronger beliefs in following medical advice. (Du et al., 2020)
Third – in the discussion of results from the multiple ordinal logistic regression model, the concept of confounding is not mentioned at all. This is what a regression model does – adjust for confounders. Thus, a factor like knowledge of TB that was clearly associated with adherence to treatment in the univariate analyses (Table 3), was not significant in the regression model with an OR of 1.001 (~1). I think it should be discussed that changes in OR of the different factors from the univariate analyses to the multivariate analysis could be ascribed to confounding, and in case in what direction.
Response: Thank you for your comment. We acknowledge the importance of addressing the role of confounding in the context of our multiple ordinal logistic regression model.
In the univariate analysis (Table 3), certain factors, such as knowledge of TB, showed significant associations with adherence to treatment. However, in the multivariate model (Table 4), the odds ratio (OR) for knowledge of TB became 1.001 (~1) and was no longer statistically significant. This change suggests that the effect of knowledge on adherence may have been confounded by other variables in the model.
For instance:
Direction of confounding: It is plausible that the initial association observed in the univariate analysis was due to the interplay of knowledge with other predictors, such as perceived stigma or health service quality. Patients with higher knowledge levels may also have had better access to health services or lower stigma, which could independently influence adherence.
Adjustment for confounding: The regression model adjusts for these overlapping effects, isolating the unique contribution of each variable to adherence. This process likely revealed that knowledge of TB, while important, did not exert an independent effect on adherence when accounting for other factors.
We add a part of discussion in the manuscript emphasizing that changes in ORs from univariate to multivariate analyses reflect the adjustment for confounders as follows:
Additionally, the changes in odds ratios (ORs) from univariate to multivariate analyses, as observed for knowledge of TB, highlight the role of confounding. In the univariate analysis, knowledge of TB showed a significant association with adherence; however, in the multivariate model, the OR became 1.001 (~1) and was no longer significant. This shift suggests that the initial association observed in the univariate analysis was confounded by other factors such as health service quality or perceived stigma. These variables may have independent effects on adherence and, when taken into account in the multivariate model, decreased the apparent impact of knowledge on adherence. This result emphasizes how TB drug adherence is multifaceted and how crucial it is to control for confounders in order to identify the distinct contributions of each variable. Understanding these dynamics emphasizes the need for tailored interventions addressing both individual knowledge and broader systemic and social barriers to adherence.
Reviewer 2 Report (Previous Reviewer 4)
Comments and Suggestions for Authors
Dear Authors;
The title is clear and informative but shortening it while remaining key elements For example eliminate of Sub-district and Pulmonary
Line 197 “A Cronbach's alpha of 0.7 or higher was considered acceptable, indicating good reliability” What was the Cronbach's alpha in your study?
In table 2 last column must be merge and centered for each variable insteat “-“
Pleas reference table 4 in section 3-4
Ensure consistency in terminology : Pleas correct M,F to male , female and Health service factor to Health service quality,..
Pleas change 0a in baseline category change to Ref in table 4
Author Response
The title is clear and informative but shortening it while remaining key elements For example eliminate of Sub-district and Pulmonary
Response: Thank you for your suggestion regarding the title. While we appreciate your feedback on shortening the title for brevity, we believe the current title effectively conveys the essential elements of the study for the following reasons:
Use of 'Pulmonary': Including the term 'Pulmonary' is important to distinguish this study from research focusing on other forms of tuberculosis, such as extrapulmonary TB. This differentiation is critical as the factors influencing medication adherence can vary significantly between pulmonary and non-pulmonary TB cases.
Inclusion of 'Dili' and Sub-district Names: While Metinaro and Becora are sub-districts, specifying them alongside 'Dili' provides a precise geographical context. This is particularly relevant given the unique socio-cultural and healthcare dynamics in these sub-districts, which are crucial to understanding the study’s findings.
We hope this explanation clarifies our reasoning for retaining the current title, which reflects the specific focus and context of the research.
Line 197 “A Cronbach's alpha of 0.7 or higher was considered acceptable, indicating good reliability” What was the Cronbach's alpha in your study?
The Cronbach's alpha in our study is shown in Result Part: Section 3.1. Content validity and reliability of the questionnaire: The questionnaire also showed acceptably internal consistency with Cronbach's alpha values of 0.78 for the Health Service-Related Factors and Community Support subscales and 0.7 for the Stigma and Family Support subscale. These findings suggest that the questionnaire is appropriate for the primary study and has sufficient reliability.
In table 2 last column must be merge and centered for each variable insteat “-“
Response: Thank you for your comment. We have merged and centered for each variable and do not use “-“
Please reference table 4 in section 3-4
Response: We have added "In Table 4, the ordinal logistic regression analysis identified health services quality..."
Ensure consistency in terminology : Please correct M,F to male , female and Health service factor to Health service quality,..
Response: Thank you for your comment. We have corrected M,F to male , female and Health service factor to Health service quality in Table 4.
Please change 0a in baseline category change to Ref in table 4
Reponse: We have changed 0a to Ref in Table 4
Reviewer 3 Report (Previous Reviewer 5)
Comments and Suggestions for Authors
With the corrections made, the article can be published.
Author Response
With the corrections made, the article can be published.
Reponse: Thank you so much for your appreciation
Reviewer 4 Report (Previous Reviewer 6)
Comments and Suggestions for Authors
The suggested changes have been introduced.
Format tables in particular font size
Author Response
The suggested changes have been introduced.
Format tables in particular font size
Response: We have formatted tables in font Palatino Linotype, size 9
This manuscript is a resubmission of an earlier submission. The following is a list of the peer review reports and author responses from that submission.
Round 1
Reviewer 1 Report
Comments and Suggestions for Authors
The aim of the paper is to identify possible factors related to (low) adherence to TB treatment in two sub-districts of Dili, Timor Leste.
This area is characterized as an area with high incidence of TB and low adherence to TB treatment. Thus, the aim of the study is justified and the research question is clear with possible large public health impacts.
However, I believe the paper is a bit premature at the moment.
First, the language is not very good and the text is difficult to follow with some flaws. As an example: Line 134: ‘The quality of the measurement.’. This sentence does not make sense.
Second, the text could be shortened substantially and focused more. As an example, the Discussion is very long and difficult to follow.
Third, many of the assessed factors are not defined in the text. Thus, the main outcome, treatment adherence, is grouped into low, medium, or high without defintions other than points on a scale, but with a reference to a paper. Such a central outcome should be defined in the paper rather than just referenced to. Also, other factors are not explained or defined, which they should, inclusive of stratification (Knowledge of PTB medication, Health service-related factors, Stigma, Family and community support).
Stratifications appear for some factors not to be quite complete: For occupation, there were four categories apart from unemployed. Were all in jobs either farmers, officers, merchants, or laborers? For income, did none earn more than 500$ per month? Urban and rural – how are those defined?
Fourth, inclusion of patients: persons under treatment for 3-6 months were included. But a cure should end after 6 months treatment. So were all participants under treatment at the time of sampling, or were also persons who had undergone treatment included?
Fifth, in the statistical section, calculation of OR and Chi-square tests are briefly mentioned as only analyses or tests, but in Table 3 it appears that a regression model is used. What model is that? Also, are each factor evaluated individually or are there adjusted for any factors? Uni or multivariate factor analyses? And if not multivariate analysis, why not do a multivariate analysis trying to identify whether of the factors age, occupation, health factor service or physical exercise (all with significant strata) matter?
Sixth, in tables it is not always clear what the percentages refer to, whether these are row or column percentages. Most appear to be column percentages, which may not always make sense alone. Thus, in Table 1 it would make sense to ask ‘how do males perform regarding adherence compared to women’, or with other words, ‘how are males distributed over the adherence groups compared to women’, which would demand row percentages. In Table 2, column percentages do not always add up to 100%, e.g. regarding the ‘Other chronic health conditions…’ factor. Titles of tables should be self-explanatory including mentioning of the study group, time and place. Some p-values seem to express tests of homogeneity for the whole factor (one p-value per factor, Tables 1 and 2), while for factors in Table 3 there are p-values for all strata. But in case of more strata and e.g. one of these strata significant, is the factor as a whole significant? Thus, in Table 3 being a merchant is significant compared to being unemployed, but does that mean that occupation as a whole is associated with the outcome? In table 3, what is the outcome? This is not mentioned. In Table 3 in ‘Side effects from TB medication’ there are 106 with an answer ‘No’, but in the line below, ‘TB medication side effects’ there are 107 with No. What is correct? Please also note that table text and figures for the factors do not always align, as in e.g. Table 3 ‘TB medication side effects’. In Table 2 footnotes mention by * different levels of significance (<0.05, <0.01, <0.001), but what does this refer to in the table? There are no ‘*’ in the tables except for the last line where there seem to miss some strata (‘Family and community support’). Also there is a footnote marked a, but what does this refer to?
Seventh, I don’t think all conclusions are justified from the results. As an example, young age is discussed as a group with low adherence to treatment. But calculated from figures in Table 1, the proportion of persons of low adherence in the young age group is 63%, while this proportion is higher in all older age groups. Also, in the low adherence group, the proportion of older persons was lower in the young age group (24.9%) but higher in the two older groups 31-50 (30%) and 51-64 (25.3%). In addition, the authors discuss risk of contraction of TB rather than adherence to treatment.
In conclusion, while the research question and aim of the study are relevant, the basic data material (questionnaire information) appears to be sound and sample size sufficient, I think that this study needs much work regarding descriptions and definitions of factors, analyses, discussion and conclusions drawn. Also, I strongly recommend a native English speaking person to do a linguistic and logical (text coherence) check of the text.
Comments on the Quality of English LanguageAs described above, the language is not very good and the text is difficult to follow with some flaws. As an example: Line 134: ‘The quality of the measurement.’. This sentence does not make sense. I strongly recommend a native English speaking person to do a linguistic and logical (text coherence) check of the text.
Author Response
The reviewer has provided edits to the background section of the manuscript (lines 49-87) and the methodology section (lines 89-195).
Additionally, the section on "Determinants of Patients’ Medication Adherence Based on Ordinal Logistic Regression" (lines 319-331) and the corresponding table of OLR results have been re-analyzed.
The discussion has also been revised and modified accordingly.
If any of the results are not relevant or if further revisions are needed, please do not hesitate to contact us at the author’s email: amentinhof06@gmail.com.

Reviewer 2 Report
Comments and Suggestions for Authors
This is a topic of great importance considering the health system and the larger benefit to the community. However, the findings have been presented as basic and need to focus on the scientific standards of this journal.

A greater section of the results, discussion and some of the concluding needs improvisation.
Author Response
The reviewer has provided edits to the background section of the manuscript (lines 49-87) and the methodology section (lines 89-195).
Additionally, English revision and the section on "Determinants of Patients’ Medication Adherence Based on Ordinal Logistic Regression" (lines 319-331) and the corresponding table of OLR results have been re-analyzed.
The discussion has also been revised and modified accordingly.
If any of the results are not relevant or if further revisions are needed, please do not hesitate to contact us at the author’s email: amentinhof06@gmail.com.

Reviewer 3 Report
Comments and Suggestions for Authors
Overall, this manuscript is technically sound. The components of the research are represented appropriately, as regard the title, introduction, methods, results, discussion, and conclusion. However, a few comments are needed to be cleared before it can be considered for publications.
Line 48 in introduction: MDR/R-R: write full words before abbreviations.
Also line 110 in methods (PTB) as it is first appearing in the text
Line 83: write day date of the study period Month/Day/ Year
Line 98-103: The authors wrote in abstract that proportional sampling was used but this wasn’t demonstrated in the methods section: please clarify total number of the two community health centers and how proportional sample was calculated for each health center.
Author Response

(The authors gave the same response as above.)

Reviewer 4 Report
Comments and Suggestions for Authors
Dear Authors;
This study entitled “ Adherence to Pulmonary Tuberculosis Medication and Associated Factors: A cross-sectional study in Metinaro and Becora Sub-district, Dili, Timor Leste” is a good topic if it is possible to reach a correct conclusion based on proper model. It seems that it is necessary to have a methodologist in the study. There are a few points that can be problematic.
None of odds ratios and their CI match each other and it is necessary to take a serious revision. odds ratios must be in the CI. based on table 3 ci maybe stated for beta and no for OR
Sentence in Line 203-208 was repeated
Line 302 The odds of lower medication adherence for patients aged between 18-30 years old are approximately 0.238 times lower than the odds of lower medication adherence for patients ager over 65 years old, (OR = 0.238, 95% CI: 0.229–2.147, P = 0.15). It s better to explained 76 % decreased
Line 134 A single sentence of ” The quality of the measurement. “ ???
Line 178 -291 Results Socio-Demographic Characteristic of study participants It is explained a lot, while the percentages and numbers can be seen in the tables. Briefly mention
Chisquare is not necessary no in text no in table
Lin 132 stated: patients into 3 groups based on the given score where a score of 8 for high adherence, a score of 6 - 7 for medium adherence, and a score of less than 6 for low adherence. In ordinary logistic that proportional odds model is an important assumption, odds ratio assessing the effect of an exposure for any of comparisons (high adherence compared to med,low Or high, med compared to low adherence) but your cut off Not specified in table 3 and text 295- 326.
In the “Determinants of Patients’ Medication Adherence Based on Ordinal Logistic Regression” section All interpretations must be corrected. The cutoff for the outcome should be clear. In this sentence “the patient with moderate medication adherence have approximately 13.464 times higher odds of being associated with health service factors compared to those with high medication adherence (OR = 306 13.464, 95% CI:1.532–3.668, P = 0.001).” should be change moderate and low or high and moderate. In this sentence “Furthermore, patients who engage in physical exercise have approximately 6.590times higher odds of medication adherence compared to those who do not engage in physical exercise (OR = 6.590, 95% CI: 1.248–2.525, P = 0.001)” what adherence? ……
A large number of variables in the model affects the significance, it is better to run the model with fewer variables.
Variables that were nonsignificant in the univariate such as Chronic health conditions )chi2 3.26 pvalue=0.515) should not be included in the model .
Discussion change based on new results of table 3
Author Response

(The authors gave the same response as above.)

Reviewer 5 Report
Comments and Suggestions for Authors
After reviewing the article, there are several observations that I would like to make:
-All the percentages in tables 1 and 2 should be reviewed, for example in the following results:
F 181 (45.5) 127 (43.3) 33 (45.2) 21 (56.6)
They should be
F 181 (45.5) 127 (70.1) 33 (18.2) 21 (11.6)
I don't know if doing this would change the statistical significance. But in the conclusions, young people have a greater adherence to treatment than older people. This result differs from other works. The authors explain how young people can spread tuberculosis but do not explain why young people have a greater adherence to treatment.
Table 3 does not show a significant result in terms of occupation, however, the conclusions state that merchant backgrounds had a significant association with medication adherence.
Therefore, the conclusions should also be revised.
Author Response

(The authors gave the same response as above.)

Reviewer 6 Report
Comments and Suggestions for Authors
This is a study on adherence to pulmonar tuberculosis medication in a region where the topic is particularly important as it has high rates of incidence of the disease and high rates of mortality from the disease. The present study does provide valuable information which may justify investment in programs to improve patient attitudes towards medication.
Title: Adequate.
Abstract: Adequate
Background: adequate
Study methodology; adequate, concern with preventing errors and bias.
A pilot study was carried out. How many participants? Were them include in the final sample? What was changed after the pilot?
Please provide a copy of the survey questions in English for peer review and for inclusion as an Appendix in the event of publication.
Is chi-square always valid? Maybe not because in contingency tables there are cells that take the value of 0 or 1.
Results: Confusing, extensive
Response rate?
This section should have a major review. There is an excess of information in the text that should be tabulated for reader-friendliness
Correct tables titles
Example:
Table 1 – Medication adherence levels related the characteristics of study participants
Table 2 – Pulmonary Tuberculosis Medication Adherence regarding …..
Table 3 – Adjusted odds ratios and 95 % CI association between Pulmonary Tuberculosis Medication Adherence and various variables.
Discussion
What can justify the results regarding the practice of physical exercises? And what about gender? You can discuss better based on relevant articles.
Conclusion – Limitations of the study must be in the discussion
References - Are current and relevant
Has a linguistic revision been carried out? A language check of a revised version of the manuscript is advisable before resubmission.
Some notes
Line 31 – chronic health condition” is repeated
Line 76 – “Therefore,….delete “in”
Line 82 - delete “whereas”
Lines 81-82/85-86 – repeated phrase
Lines 48/90 - Explain abbreviation
Line 91 – “undergoing psycotherapy” - Why? It could be interessant.
Line 134 – Delete “The quality of the measurement”
Line 180 – “.” or “,”?
Line 185 – Include “(Table 1)”
Lines 183-193 – These are just socio-demographic data, they do not refer to differences – Correct sentence in line 185.
Laborer – Who is included?
Lines 192-193 - Are montly income data correct? In the table 1?
Lines 193-194 – The sentence does not represent what is written in the table. P is not less than 0.05. It is assumed that the significance level considered was 0.05. This is confusing. Rewriting the sentence.
Linha 238 – What are “participant' perceptions”
Lines 226-241 – The data from the scales should be presented in a table. It would make reading easier Data are very extensive.
Lines - 290-291 – “no statistic computed” - table 2.
Table and text have to be greatly improved. Example: “experienced positive impact” is not the same as “ Other chronic health conditions affect respondente ability to manage TB”. The text must be coherent.
Lines 242-358 – Data presented in a confusing way.
Linha 344 - ” …age was statistically significant in medication adherence….”. Correct
“The differences resulting from the association between adherence to therapy and age are statically significant (or have statistical significance).
Line 391-398 – Correct language
Format bibliography.
Reis, J. d. (2016)
ELSA ZERBINI, A. G., SILVIA ESTRADA. (2017
Author Response

(The authors gave the same response as above.)

Reviewer 7 Report
Comments and Suggestions for Authors
The study is an important piece of scientific facts to eradicate and control such a serious disease spread and would help in national policy decision-making regarding PTB dissemination, however, I wonder why I see fewer references of recent years (2023 and 2024) in the manuscript.
The introduction mentions the issue and related facts. Tuberculosis has been one of the major diseases in Asia. The statistical approach was good and the instrument was validated appropriately.
Although the healthcare providers' roles were studied and discussed in the study, the institutional role in enrolling the PTB patients and employing DOTs was not addressed sufficiently.
Overall, the study is worth reading, is important in its content, and would be valuable in addressing the healthcare issue.
Comments on the Quality of English LanguageIn some places, English grammatical composition needs to be corrected. Line 334, influence should be "influenced". In the manuscript, there was too much use of we, our, and us pronouns that should be lessened.
Author Response

(The authors gave the same response as above.)
